# Biocontrol Potential of Chitin and Chitosan Extracted from Black Soldier Fly Pupal Exuviae against Bacterial Wilt of Tomato

**DOI:** 10.3390/microorganisms10010165

**Published:** 2022-01-13

**Authors:** Violah Jepkogei Kemboi, Carolyne Kipkoech, Moses Njire, Samuel Were, Mevin Kiprotich Lagat, Francis Ndwiga, John Mwibanda Wesonga, Chrysantus Mbi Tanga

**Affiliations:** 1Department of Botany, Jomo Kenyatta University of Agriculture and Technology, Nairobi P.O. Box 62000-02000, Kenya; violahkemboi42@gmail.com (V.J.K.); mnjire@jkuat.ac.ke (M.N.); samaringo@gmail.com (S.W.); mlagat237@gmail.com (M.K.L.); irerifin@gmail.com (F.N.); 2Department of Food and Nutritional Sciences, Jomo Kenyatta University of Agriculture and Technology, Nairobi P.O. Box 62000-02000, Kenya; 3Department of Horticulture and Food Security, Jomo Kenyatta University of Agriculture and Technology, Nairobi P.O. Box 62000-02000, Kenya; jwesonga@agr.jkuat.ac.ke; 4International Centre of Insect Physiology and Ecology (ICIPE), Nairobi P.O. Box 30772-00100, Kenya; ctanga@icipe.org

**Keywords:** tomato bacterial wilt disease, insect pupal shell, organic soil amendment, integrated pest management, *Ralstonia solanacearum*

## Abstract

Globally, *Ralstonia solanacearum* (Smith) is ranked one of the most destructive bacterial pathogens inducing rapid and fatal wilting symptoms on tomatoes. Yield losses on tomatoes vary from 0 to 91% and most control measures are unaffordable to resource-poor farmers. This study investigated the antimicrobial activities of chitin and chitosan extracted from black soldier fly (BSF) pupal exuviae against *R. solanacearum*. Morphological, biochemical, and molecular techniques were used to isolate and characterize *R. solanacearum* for in vitro pathogenicity test using disc diffusion technique. Our results revealed that BSF chitosan significantly inhibited the growth of *R. solanacearum* when compared to treatments without chitosan. However, there was no significant difference in the antibacterial activities between BSF and commercial chitosan against *R. solanacearum*. Soil amended with BSF-chitin and chitosan demonstrated a reduction in bacterial wilt disease incidence by 30.31% and 34.95%, respectively. Whereas, disease severity was reduced by 22.57% and 23.66%, when inoculated tomato plants were subjected to soil amended with BSF chitin and chitosan, respectively. These findings have demonstrated that BSF pupal shells are an attractive renewable raw material for the recovery of valuable products (chitin and chitosan) with promising ability as a new type of eco-friendly control measure against bacterial wilt caused by *R*. *solanacearum*. Further studies should explore integrated pest management options that integrate multiple components including insect-based chitin and chitosan to manage bacterial wilt diseases, contributing significantly to increased tomato production worldwide.

## 1. Introduction

The bacteria *Ralstonia solanacearum* (Smith) is a soil-borne plant pathogen that affects mainly solanaceous plants and some non-solanaceous plants [1]. They are Gram-negative, short rod, strict aerobe, non-spore-forming, motile bacteria. Molecular characterization of *R. solanacearum* strains based on sequence analysis of internal spacer (ITS) region has resulted in four phylotypes divisions [2]. *Ralstonia solanacearum* has been reported to be the causal agent of bacterial wilt disease in over 200 host plant species belonging to 50 plant families [3]. These bacteria infect plants from planting media through root injuries caused either by mechanical, insect, or nematode damage [4]. The bacteria has been reported to colonize the xylem vessels of the plants leading to massive destruction through the blockage of the water transportation system [4]. This causes the plant to develop characteristic wilting signs and symptoms that include yellowing of leaves, stunted plant growth, and sudden death [5]. This characteristic dead process caused by the *R. solanacearum* is referred to as vascular wilt disease and the disease has been ranked as the second most important bacterial pathogen worldwide [6].

Remarkably, it has been widely documented as one of the most destructive pathogens identified to date due to its ability to induce rapid and fatal wilting symptoms in infected host plant species with severe economic impact [7]. However, the direct yield losses caused by *R. solanacearum* vary widely according to the host, cultivar, climate, soil type, cropping pattern, and strain [6]. Previous studies have highlighted yield losses ranging between 0 and 91%, 33–90%, 10–30%, 80–100%, and up to 20% for tomato, potato, tobacco, banana, and groundnut, respectively [6,8]. Many studies have associated the difficulties of controlling *R. solanacearum* with its ability to grow endophytically, survive in soil, especially in the deeper layers, travel along the water, and its relationship with weeds [6]. This observation is consistent with the report by [9], who reported the management of *R. solanacearum* through the use of disease-resistant cultivars, crop rotation, and field sanitation have been largely unsuccessful.

Although synthetic bactericides have shown success, they are associated with air, underground water, soil pollution, residual toxicity effects [10], loss of biodiversity [11], and development of pathogen resistance [12]. Thus, there has been increased interest in the use of alternative management strategies for *R. solanacearum* including the use of potential biological control agents such as antagonistic microorganisms [13]. However, the use of these microorganisms is associated with challenges relating to storage time [14]. There is, therefore, a need for the development of alternative control agents such as plant extracts [15] and biopolymers such as chitin and its derivatives [16] for use in the management of plant disease. Chitin is a natural polysaccharide occurring as a structural component of living organisms such as insects, fungi, and crustaceans [17]. Chitosan refers to an N-acetyl D-glucosamine polysaccharide usually obtained through deacetylation of chitin using sodium hydroxide or chitinase enzymes. In agriculture, chitin derivatives are ranked first in the list of basic compounds approved in the European Union (EU) for their application in organic agriculture and integrated pest management systems [18]. Chitosan has unique biological properties such as biodegradability, biocompatibility, non-toxicity, and antimicrobial activities [19]. Due to these advantages, they have been applied in various fields such as in the food industry, agriculture, medicine, wastewater treatment, paper, and cosmetics industries [20]. Conventionally, organisms such as crabs, shrimps, krill, lobsters, and waste from the seafood industry have been used as the commercial source of chitin. Additionally, research has shown that insects such as beetles, grasshopper [21], and black soldier fly can be a potential source of chitin production.

Black soldier fly (BSF; *Hermetia illucens*) is a known resource insect native to the tropical regions of America and South America but recently it is distributed in almost all parts of the world. This fly feeds naturally in decaying organic matter and the process converts biomass nutrients into its biomass thus it constituent a means of recycling waste in the environment. BSF has also been shown as a valuable source of proteins, lipids, fats, and other minerals. Due to this, BSF rearing technology is currently increasing mainly for the production of poultry, pig and fish feeds, and biodiesel. Additionally, BSF has been reported as a potential source of chitin. Although chitin and chitosan from black soldier fly have been shown to have similar properties to those from other sources, there is no information and research attention on their antibacterial activities against severe plant diseases such as tomato bacterial wilt. Hence, this study aimed at investigating the antimicrobial activities of chitin and chitosan extracted from black soldier fly pupal exuviae waste against *R. solanacearum* to guide their introduction into the integrated pest management toolbox to manage bacterial wilt diseases and contribute to increased tomato production worldwide.

## 2. Materials and Methods

### 2.1. Study Site

This study was conducted at Jomo Kenyatta University of Agriculture and Technology (JKUAT), Juja Kiambu County (1.0891 °S, 37.0105 °E, altitude of 1400 asl). In vitro and field tests were done at the Department of Botany laboratory and Department of Horticulture and Food Security greenhouse, respectively. The pupal exuviae from which both chitin and chitosan were extracted was obtained from the BSF mass production facility at the Animal Rearing and Containment Unit (ARCU) at the International Centre of Insect Physiology and Ecology (*icipe*), Nairobi, Kenya.

### 2.2. Soil Sample Collection and Isolation of Ralstonia solanacearum

Soil samples were randomly collected from tomato growing fields that had dead tomato plants and those depicting wilting symptoms at Jomo Kenyatta University of Agriculture and Technology (JKUAT). Blocks were earmarked for the trial then divided into plots. Three independent plots were selected then subdivided into numbered units. Three units per plot were randomly selected and soils collected (in triplicates) by digging around the plant roots at 30 cm depth using a sterile jembe. Ten grams of the sampled soil was suspended in 100 mL of sterile distilled water then agitated for 10 min on a mechanical shaker. Serial dilution was performed by transferring 1 mL of the suspension into 9 mL of sterile distilled water in test tubes to dilution factor 10^−6^. Plating was done by pipetting and spreading 0.1 mL of the suspension from each dilution tube onto sterile Triphenyl tetrazolium chloride (TZC) agar plates. The plates were then incubated at 37 °C for 24 h after which bacterial growth was measured by counting colonies of *R. solanacearum* and documented.

### 2.3. Cultural Characterization

Typical cultural characteristics of *R. solanacearum* were determined using 24 h old cultures grown at 37 °C on TZC gar according to the procedure done by [22]. The fluidal, irregularly round colonies with white margins and light pink centers were considered to be *R. solanacearum* colonies [23,24] and were sub-cultured to obtain pure cultures for subsequent use.

#### 2.3.1. Morphological Characterization

Gram staining of pure cultures of the *R. solanacearum* was done following the procedure described by [25]. This involved spreading a loopful of the colonies on a glass slide and heat-fixing on a very low flame. Aqueous crystal violet solution was spread over the smear and left for 30 s then washed with running tap water for one minute. It was then flooded with iodine and left for one minute, rinsed with tap water, then decolorized with 95% ethanol until clear runoff. After washing, the smear was counter-stained with safranin for 30 s, washed with tap water, dried, and observed at a magnification at ×100 using oil immersion. Pink short rod-shaped cells were considered a positive test for *R. solanacearum* [26].

#### 2.3.2. Biochemical Characterization

The bacterial solubility in 3% KOH was examined to eliminate any possible confusion of the bacterial pathogens that cause wilting in tomatoes as described by Khasabulli et al. (2017). The pure culture of the pathogen was picked using a sterile wire loop and placed on the glass slide containing a drop of 3% KOH solution. It was stirred for about 10 s then raised for a few centimeters from the slide while observing for mucoid/slime threads. The formation of a viscous solution or slime thread indicated a positive test (KOH soluble). A catalase test was done as described by Khasabulli et al. (2017), whereby a loopful of fresh bacterial culture of *R. solanacearum* was mixed with a drop of 3% hydrogen peroxide (H_2_O_2_) on a glass slide and observed for the production of gas bubbles. An observation of effervescence suggested a positive test [23,25].

A gas production test for the pure cultures of isolated *R. solanacearum* was conducted according to the procedure done by Pawaskar et al. (2014). Sterile nutrient broth with 2% glucose in test tubes containing inverted Durham tubes was used. The test tubes were inoculated with 0.5 mL of bacterial suspension and incubated at 37 °C for 24 h. The presence of air bubbles in the inverted Durham tube was an indication of gas production hence considered a positive test. The ability of *R. solanacearum* isolate to hydrolyze starch in nutrient agar was tested according to the procedure done by Pawaskar et al. (2014). The bacterial cultures of *R. solanacearum* were inoculated and spread on the center of sterile nutrient agar plates containing 0.02% starch then incubated at 37 °C for 24 h. After incubating, the plates were flooded with Lugol’s iodine. A clear zone around bacterial culture suggested a positive test [25,27].

#### 2.3.3. Molecular Characterization

The genomic DNA was extracted using Quick-DNA^TM^ Fungal/Bacterial Miniprep kit (Zymo Research Corp, Irvine, CA, USA). A pure culture of *R. solanacearum* was sub-cultured on Casamino acids Peptone Glucose (CPGA) plates and incubated overnight then used for extraction of DNA based on the manufacturer’s instructions. The extracted DNA was run under 1% agarose gel electrophoresis and visualized on an ultraviolet Transilluminator (UVT-20 SML model). Multiplex PCR was performed for phylotype identification and confirmation of the isolated *R. solanacearum* diseased plants from JKUAT fields using phylotype-specific primers as described by Fegan and Prior (2005). The reaction was carried out in a total volume of 25 µL reaction containing: One Taq 2X Master mix, phylotype-specific forward primers; Nmult:21:1F, Nmult:21:2F, Nmult:22: lnF, Nmult:23:AF, reverse primer Nmult:22:RR, DNA template, and Taq DNA polymerase. Amplifications were done in an Eppendorf AG, 22331 Hamburg, Germany thermocycler using the following conditions: initial denaturation at 94 °C for 3 min, 35 cycles of denaturation at 94 °C for 30 s, annealing at 55 °C for 30 s, extension at 72 °C for 30 s, and a final extension step at 72 °C for 7 min. The amplified PCR products were run through electrophoresis on 1% (*w/v*) agarose gel and visualized on a UV transilluminator [28].

### 2.4. Pathogenicity Test

A pathogenicity test was conducted on one-month-old healthy tomato seedlings (Prostar F1) using *R. solanacearum* from the International Centre of Potato Research (CIP) and those isolated from JKUAT fields. The soil drench method was used for inoculation of the plants with the pathogen according to the procedure by Kariuki (2020). The seedlings were transplanted onto plastic pots containing autoclaved cocopeat mixed with Hoagland’s solution. The plants were left in the medium for three days to let them stabilize. After three days of stabilization in the medium, the seedlings were injured on the stems just above the growth media surface and the roots using a sterile scalpel blade. Turbidity of bacterial suspension was adjusted to 0.5 McFarland equivalents then 10 mL drenched on the injured stems and roots using a sterile syringe. A set of transplanted seedlings drenched with sterile distilled water served as the control. Each treatment consisted of 10 plants one per pot and the experiment was done in triplicates. The plants were then monitored for bacterial wilt symptoms in the greenhouse under natural sunlight conditions. Plants with at least one wilted leaf were considered diseased. After the development and appearance of wilting symptoms, the pathogen was reisolated on TZC media and re-identified using cultural, morphological, biochemical, and molecular methods [29].

### 2.5. Extraction of Chitin and Chitosan

Chitin and chitosan were extracted from the BSF pupal exuviae using chemical methods as described by Kaya et al. (2015). The pupal exuviae were sorted, washed thoroughly with tap water, and rinsed twice with distilled water, sun-dried, and ground into a fine powder using a blender. The obtained pupal exuviae powder was first demineralized by treating 100 g of the powder with 1000 mL of 1 M hydrochloric acid then boiled for two hours to remove all the minerals. The demineralized material was then washed with distilled water until a neutral pH was achieved, then filtered and dried in an oven at 60 °C for 6 h. The dried demineralized powder weighing 100 g was then treated with 1000 mL of 1 M sodium hydroxide (NaOH) then boiled for 4 h to remove proteins. The extracted chitin material was washed thoroughly with distilled water until it attained a neutral pH and dried in an oven at 60 °C for 6 h. Thereafter, the chitin was further processed to produce chitosan as described by Kaya et al. (2015) but with slight variations. One hundred grams of dried chitin was refluxed in 1000 mL of 40% NaOH then boiled for 8 h with continuous stirring. The obtained product was washed thoroughly with distilled water to a neutral pH then dried in an oven at 60 °C. Chitosan (CHT) extracts weighing 0.5, 1, 2.5, and 5 g were dissolved in 100 mL of 1% (*v/v*) acetic acid solution separately to obtain different concentrations of CHT (*w/v*). Following the dissolution of chitosan in acetic acid, it was sterilized by autoclaving at 121 °C for 15 min and stored at 4 °C for later use [30,31].

### 2.6. Antimicrobial Effects of BSF Chitosan on Ralstonia solanacearum

Antimicrobial activities of BSF chitosan against *R. solanacearum* were tested using disc diffusion [32]. Pure cultures of *R. solanacearum* were sub-cultured overnight at 37 °C on casamino peptone glucose (CPG) agar media. Bacterial colonies from the plates were suspended in sterile normal saline and adjusted to obtain an equivalent of 0.5 McFarland. then pipetted onto 20 mL of Mueller Hilton Agar plates. L-form glass spreader was sterilized by passing over a flame and used to spread the bacterial suspension over the Agar plates. Sterile Whatman’s filter paper discs of 6 mm diameter were soaked in different concentrations (0, 0.5, 1, 2.5, and 5%) of sterilized BSF chitosan. Chitosan-soaked discs were air-dried then gently picked using sterile forceps and placed on the surface of inoculated agar plates at equidistant positions. Filter paper discs soaked in sterile distilled water were used as the negative control. The plates were incubated at 37 °C for 24 h The inhibition zones were measured in millimeters and average inhibition zones were calculated for the three replicates. The “control treatment” consisted of different concentrations of commercial chitosan (0, 0.5, 1, 2.5, and 5%) and eight different antibiotic discs: Ampicillin (25 µg), Tetracycline (100 µg), Nitrofurantoin (200 mcg), Nalidixic acid (30 µg), Streptomycin (25 µg), Sulphamethoxazole (200 µg), and Cotrimoxazole (25 µg) [33].

### 2.7. Effects of BSF Chitosan on Ralstonia solanacearum Symptom Expression in Tomato

Tomato seeds (Prostar F1 from Simlaw Seeds Company, Nairobi, Kenya) were grown in seed propagating units for 30 days. The seedlings were then transplanted onto plastic pots containing 200 g of sterile growth media (cocopeat) and 20 g of BSF-derived chitin and chitosan separately. Three days after transplanting 50 mL of Hoagland’s solution was added to the transplanted plants. Fourteen days after transplanting, the plants were inoculated by drenching 10 mL of 0.5 McFarland suspension of *R. solanacearum*. The plants were injured using sterile scalpel around roots and on stem slightly above the growth media surface. The pathogen was drenched around the injured surface using a sterile syringe. Seedlings transplanted into sterile growth media devoid of chitin or chitosan extracts and inoculated with *R. solanacearum* served as the positive control. Other seedlings transplanted onto growth media without chitin or chitosan and not inoculated with *R. solanacearum* bacterial suspension served as the negative control. The experiments were replicated three times and arranged in a completely randomized design in the greenhouse under natural light conditions. The plants were irrigated after every 12 h and other standard agronomic practices were observed to ensure that plants were free from any form of stress.

Bacterial wilt disease incidence (DI) was monitored daily for 1 month after pathogen inoculation. Plants with wilted leaves were recorded as diseased plants. The percent disease incidence was calculated using the formula by Kempe and Sequeira (1983):$$\%\;\mathrm{DI}=\frac{n}{N}\times 100$$
where *n* = Number of wilted leaves per plant, *N* = total number of leaves.


Bacterial wilt disease severity was evaluated based on the scale (0–5) of Kempe and Sequeira (1983) where 0 = no symptoms; 1 = 1–25% leaves wilted; 2 = 26–50% leaves wilted; 3 = 51–75% leaves wilted; 4 = more than 75% but less than 100% of leaves wilted; 5 = all leaves wilted and plant death.

The percent disease severity (DS) was calculated using the formula [34]:% DS = [Σ (ni × vi) ÷ (V × N)]
where ni = number of plants with the respective disease rating; vi = disease rating; V = the highest disease rating; and N = the number of plants observed

### 2.8. Data Collection and Analysis

Data on cultural, biochemical, and molecular characterization were collected by observation, and the results were recorded on excel sheets. The data on antimicrobial activity was collected by measuring clear zones of inhibition around each disc on each plate of each test organism. All experimental data were expressed as means ± standard error. Data analysis was done using Stata SE-64 2011 statistics software and means separated using the Bonferroni range test. The difference in mean inhibition zone was done using a one-way analysis of variance, and the difference was considered significant at *p* ≤ 0.05.

## 3. Results and Discussion

### 3.1. Characterization of Ralstonia solanacearum

Kelzman’s tetrazolium chloride (TZC) media-enabled successful isolation of *R. solanacearum* from the soil sample collected from the JKUAT farm and differentiation of virulent and avirulent strains. Most colonies on the observed plates appeared as large, fluidal irregularly round white with a pale red to the pink center on TZC media hence as per Balabel (2018) they were virulent [35] (Figure 1).

The observation of Gram staining reaction done on the colonies of *R. solanacearum* isolates showed pink small rod-shaped cells under the microscope at ×100 magnification (Table 1). These Gram staining reaction observations are in line with those reported by Mutimawurugo et al. (2019) where all isolated *R. solanacearum* from three regions in Rwanda were Gram-negative short rods. This is because *R. solanacearum* is a Gram-negative rod-shaped bacterium and that all plant pathogenic bacteria are usually Gram-negative except *Streptomyces* and *Clavibacter* [23,36].

There was a formation of viscous thread following KOH tests on all the isolates. The observations conform to that reported by Teli et al. (2018) where the formation of mucoid or slime thread was observed when the bacterial suspension of *R. solanacearum* was raised from a glass slide containing 3% KOH [9]. The formation of viscous, mucoid, or slime thread is due to the outer membrane of Gram-negative bacteria being readily disrupted when exposed to 3% KOH releasing the viscous DNA [23]. The potassium hydroxide solubility test is an important test for confirmation of the Gram-negative test and helps in differentiating *R. solanacearum* from other wilt-causing microorganisms such as *Fusarium* [22].

The catalase test showed that all the isolates were able to produce gas bubbles when they were mixed with a drop of 3% hydrogen peroxide hence were considered as catalase positive. These findings conform to those found by Khasabulli et al. (2017), whereby *R. solanacearum* isolated from the Maseno region was catalase-positive [23]. The gas production mixed with hydrogen peroxide is due to the ability of *R. solanacearum* to produce catalase enzyme. Catalase enzyme protects the bacteria from the toxic effects of hydrogen peroxide by catalyzing its breakdown into water and oxygen [37]. The production of gas bubbles also shows that the bacteria are aerobic or facultative anaerobe. All isolates of *R. solanacearum* were able to produce gas from glucose after 24 h of incubation hence was considered positive for the gas test. This was also observed by Khasabulli et al. (2017), when they experimented on *R. solanacearum*, they found that *R. solanacearum* was able to produce gas within 18 h of incubation. The isolates under investigation were not able to hydrolyze starch. This was a result of the absence of clear zones on the agar after incubation, confirming that the isolates were for *R. solanacearum.* The findings were similar to the results of the studies conducted by Sharma and Singh (2019) and Bawari and Narendrappa (2019) who found that *R. solanacearum* was unable to hydrolyze starch [22,24].

Based on the molecular characterization of the samples, it was clear that the bands observed were in the range of 372 bp, confirming that the sample was *R. solanacearum* (Figure 2). The findings are concurring with that of Paudel et al. (2020) who observed that *R. solanacearum* has four distinct phylotypes. Phylotype I generates 144 bp for primers Nmult21:1F/Nmult22: RR, phylotype II generates 372 bp for primers Nmult21:2F/Nmult22: RR, phylotype III generates 91 bp for primers Nmult23: AF/Nmult22: RR phylotype IV generates 213 bp for primers Nmult22:InF/Nmult22:RR [2,38]. Based on phylotype groups, the *R. solanacearum* species complex is divided into subspecies for instance phylotype I and phylotype III are classified as *R. pseudosolanacearum* phylotype II as *R. solanacearum* and phylotype IV as *Ralstonia syzygii* [39]. Therefore, the findings of this study confirm that phylotype II, which is *R. solanacearum* species is present in the JKUAT farm.

### 3.2. Pathogenicity Test

The *R. solanacearum* was confirmed to cause bacterial wilt in tomato plants. Bacterial wilt symptoms including yellowing of the younger lower leaves, dropping of the leaves, and wilting of the plants’ leaves and stem during daytime were observed two weeks after transplanting and inoculation (Figure 3). One month later, there was the formation of brown discoloration of vascular bundles in the stem above the soil layer. Additionally, the infected plants showed milky white substance streaming from the cross-sectional cuttings of the stem a sign of bacteria in the vascular tissues [23].

Diameters of zones of inhibitions produced by BSF-based pupal exuviae chitosan and commercial chitosan showed no statistically significant differences. However, there were statistically significant differences in the antibacterial activities against the *R. solanacearum* among the various treatments (Table 2).

The exact mechanisms of antimicrobial activities of BSF-based pupal exuviae chitosan are not well known. However, there are some proposed modes of action reported in previous studies. The most known mechanism involved in antimicrobial activities is attributed to the presence of amino groups on the chitosan structure that causes electrostatic interaction between their positive charges and negative charges of lipopolysaccharide on the bacterial cell membrane [40]. The electrostatic interaction has been known to disrupt the cell wall and cell membrane thus interfering with its permeability. Another postulate is that chitosan anionic charges (amino group) bind to the negatively charged phosphate groups on DNA and amino acids of the proteins thus inactivating or inhibiting messenger RNA (mRNA) and protein synthesis [41]. Chitosan can also inhibit bacterial growth by chelating essential nutrients and metals [42].

Tomato plants treated with BSF-based pupal exuviae chitin and chitosan showed a reduction in disease incidence by 30.31% and 34.95%, respectively, whereas the disease severity was reduced by 22.57% and 23.66%, respectively, compared to the control (Table 3).

These findings are in agreement with those reported by Algam et al. (2016) where they revealed the application of chitosan from shrimps through soil drenching reduced wilting incidence by 72% in tomatoes. Similarly, chitin and chitosan-based polymers showed a reduction in disease severity in cabbage and strawberry plants after being challenged with *Alternaria brassicicola* and *Colletotrichum fructicola* in a study carried out by Parada et al. (2018), reduced incidence of *Fusarium* wilt disease in tomato plants (Malerba & Cerana, 2018), and enhanced defense against *Meloidogyne javanica* in tomato plants [14,43,44,45]. Tomato plants uninoculated with *R. solanacearum* did not show wilting symptoms (Figure 4). There was no significant difference in bacterial wilt disease incidence and severity between BSF-based pupal exuviae chitin and chitosan.

The reduction of wilting disease incidence and severity by BSF-based pupal exuviae chitin and chitosan is attributed to proposed mechanisms that include direct antibacterial activity through cell lysis [41], and the formation of a mechanical barrier that protects against pathogens [46]. They can trigger plant defense responses and activate different pathways which increase the plant’s resistance to diseases [12]. In certain instances, they can stimulate hypersensitive responses in plants primarily around the infection site that results in programmed cell death. The hypersensitive response can be followed by systemic responses that include the synthesis and build-up of pathogenesis-related proteins, phytoalexins, and phenolic compounds. The systemic response can also modulate the activity of key enzymes such as chitinase, catalase, and superoxide dismutase peroxides that are involved in metabolic pathways in the defense response [41]. Chitosan showed better results than chitin which may have been caused by the deacetylation and solubility of chitosan. Recent studies have shown chitosan’s ability to increase photosynthesis, plant tolerance to stress, and expression of defense genes which helps the plant to better fight infection [47].

## 4. Conclusions

The early detection of *R. solanacearum* belonging to phylotype II is essential for preventing its introduction into new areas due to its ability to survive for prolonged periods in soil, water, and plant materials. The avoidance of tomato losses due to this pathogen would significantly contribute to increased crop production in the country. Although, many researchers have managed bacterial wilt with biological, physical, and chemical methods and/or with cultural practices; this study provides the first evidence on the efficiency of BSF-based pupal exuviae chitin and chitosan antimicrobial activities and potential to protect against *R. solanacearum* in tomato plants. The remarkable reduction of bacterial wilt disease incidence by 30.31% and 34.95% in cocopeat amended with chitin and chitosan, respectively, isolated from BSF pupal exuviae demonstrated these shells are an attractive renewable raw material for the recovery of valuable product with promising ability as a new type of eco-friendly control measure against bacterial wilt, *R*. *solanacearum*. From the findings of this study, chitosan showed better results than chitin hence use of chitosan is the best approach. Further studies should explore the effects of direct application of the BSF pupal exuviae on *R. solanacearum* in soil and their integration into integrated pest management strategies that involve multiple components to manage bacterial wilt diseases, thus significantly contributing to increased tomato production worldwide. Our primary goal is to contribute to environmentally safe, sustainable, and increased tomato production for income and food security. However, research attention to cost–benefit analyses is indispensable in the short, middle, and long term.

## Figures and Tables

**Figure 1 microorganisms-10-00165-f001:**
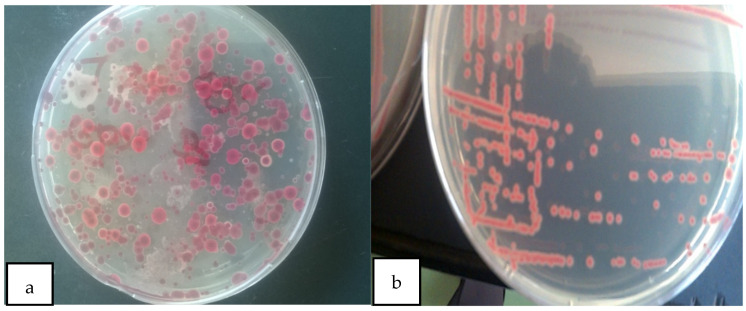
(**a**) Mixed colonies of virulent and avirulent *R. solanacearum*; (**b**) pure colonies of virulent *R solanacearum*.

**Figure 2 microorganisms-10-00165-f002:**
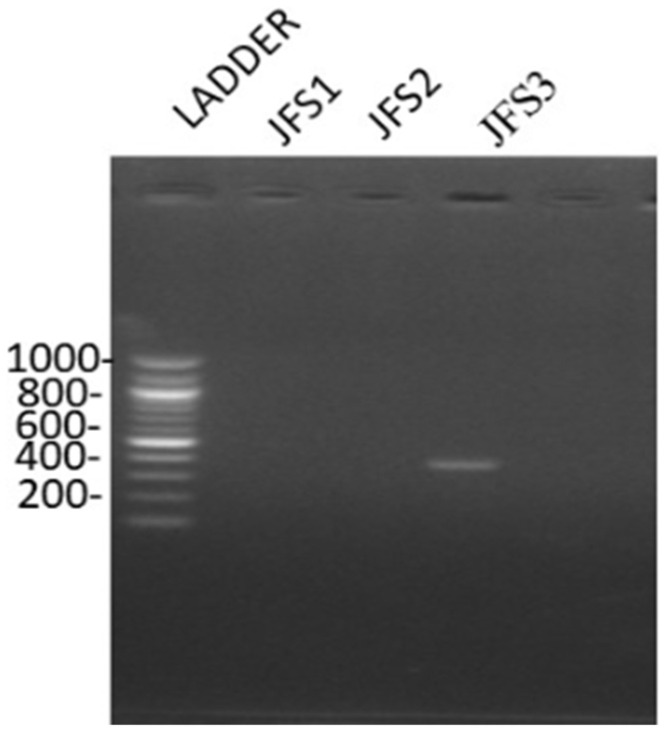
*R. solanacearum* DNA bands as observed under UV-transilluminator.

**Figure 3 microorganisms-10-00165-f003:**
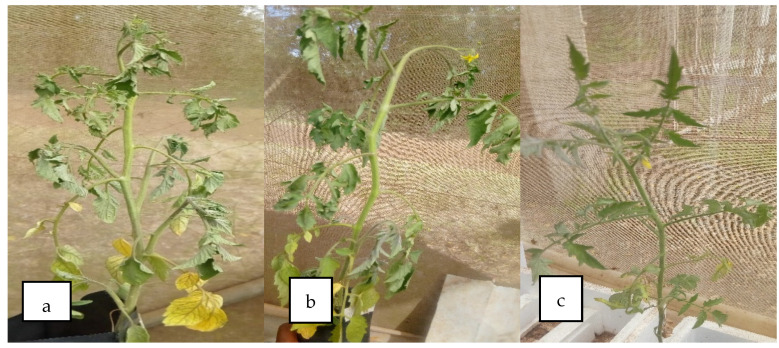
Tomato plants showing bacterial wilting symptoms: (**a**) plant inoculated with *R. solanacearum* isolated from JKUAT farm, (**b**) plant inoculated with *R. solanacearum* obtained from International Potato Centre (CIP), and (**c**) plant not inoculated.

**Figure 4 microorganisms-10-00165-f004:**
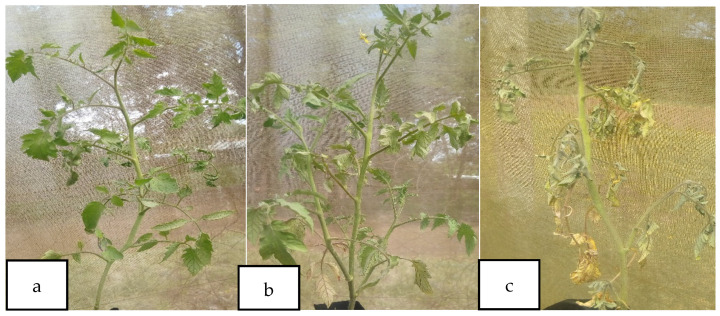
Wilting symptoms in (**a**) BSF-based pupal exuviae chitosan treated plant, (**b**) BSF-based pupal exuviae chitin treated plant, and (**c**) non-treated plant.

**Table 1 microorganisms-10-00165-t001:** Gram staining and biochemical reaction tests for the isolated *R. solanacearum*.

Test	Observation
Gram stain reaction test	−
Potassium Hydroxide solubility test	+
Catalase test	+
Gas production test	+
Starch hydrolysis test	−

− (negative), + (positive).

**Table 2 microorganisms-10-00165-t002:** Antimicrobial effects of BSF, commercial chitosan, and antibiotic discs against *R. solanacearum*.

Treatment	Means of Inhibition (mm ± Standard Error of the Mean)	*p*-Value
Sterile distilled water	0 ± 2.35 ^d^	<0.001
1% acetic acid without chitosan	11.67 ± 2.35 ^b^	0.002
BSF chitosan	19.83 ± 1.17 ^a^	0.004
Commercial chitosan	18.5 ± 1.17 ^a^	<0.001
Nalidixic acid (30 µg)	26 ± 2.349153 ^e^	0.002
Streptomycin (25 µg)	10.67 ± 2.35 ^b^	0.041
Sulphamethoxazole (200 µg)	16.67 ± 2.35 ^c^	0.021
Cotrimoxazole (25 µg)	21 ± 2.35 ^c^	0.021 <0.001
Gentamycin (10 µg)	20 ± 2.35 ^c^	<0.001
Ampicillin (25 µg)	0 ± 2.34 ^d^	<0.001
Tetracycline (100 µg)	20 ± 2.34 ^c^	0.002
Nitrofurantoin (200 µg)	15.67 ± 2.34 ^c^	0.002

Data shown are means of three replications. Means with different letters (superscript) are significantly different at *p* ≤ 0.05.

**Table 3 microorganisms-10-00165-t003:** Bacterial wilt disease incidence (DI) and disease severity (DS).

Treatment	Mean % Disease Incidence ± Standard Error of the Mean	Mean % Disease Severity ± Standard Error of Mean
BSF-based pupal exuviae chitin	30.31 ±1.36 ^a^	22.57 ± 2.01 ^a^
BSF-based pupal exuviae chitosan	34.95 ± 0.60 ^a^	23.66 ± 1.15 ^a^
Positive control	44.78 ± 1.56 ^b^	36.95 ± 1.49 ^b^
*p*-values	<0.001	<0.001

Data shown are means of three replications. Means in a column followed by the same letter are not significantly different at *p* ≤ 0.05.

## Data Availability

Not applicable.

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
