# Peer review of "Biocontrol Potential of Chitin and Chitosan Extracted from Black Soldier Fly Pupal Exuviae against Bacterial Wilt of Tomato"

_microorganisms, 2022, doi:10.3390/microorganisms10010165_

Round 1

Reviewer 1 Report

Please see attached file for detailed comments.

Author Response

In general, the paper is well written and the data collection described was thorough and included sufficient details for the replication of this work to be achieved. Detailed below are requests for clarification in particular areas and suggestions for figure improvements. The targeted journal for this work raises some concerns about the paper in its current form. There are areas that I suggest to elaborate on in order to help the audience better understand the implications of the work.

This has been introduction in the first section of the paper line 83

The first suggestion that I strongly recommend is to take some time in the introduction of the paper to describe the black soldier fly industry. As written, the authors argue that the chitin resource being explored is sustainable. It is assumed that the readers would be aware of the process and intent to rearing this insect in vast quantities. This is not true. The industry and process needs to be introduced.

This is introduced in line 90-97

The second suggestion is to provide more details regarding the mechanisms at work as chitin interacts with the plant to strengthen its defenses. Is the pathway known? Does the chitin have to be extracted and purified to solicit the response or can you add the pupae, intact, to the soil and achieve the same results? If this information is NOT known, I would like to see the authors mention this as areas for future research.

Highlighted in line 441

This is emphasized below but the authors NEED to provide more details on the data collection and analysis. Statistical tables presenting the data analysis results (i.e. ANOVA tables) are highly recommended. The number of replicates and analysis are not clear for most, if not all, of the work presented.

Specific line edits and suggestions follow:

Line 227-229: should read “The plants were injured using a sterile scalpel around the roots and on the stem slightly above the growth media surface. The pathogen was…”

corrected

Line 246: period missing at the end of the sentence

corrected

Section 2.8: More information is needed in regards to the culture evaluations. Did you measure the area of infection, or presence/absence? How many replicates were conducted? When reporting the statistical results you need to include the degrees of freedom and the f-statistic; the p-value alone is insufficient. This can be written in the text or provided in a table.

F statistic was not computed since there was no data modelling done. The number of replicates has been indicated in the table

Line 267: were should be where corrected

Line 294: should read “This was a result of…” corrected

Line 319: remove the coma after substance and stem

Figure 4 is completely ineffective. The sections on the various plates need to be labeled. Is there a positive control? What is being evaluated and how (goes back to the above question about the area or presence/absence of pathogen)? This is not clear.

The figure has been removed

Line 341: previous studies. The most known mechanism… corrected

Line 360: that should be the corrected

Figure 5 legend: remove is a from part (c) corrected

Line 384: chitosan. Recent corrected

Reviewer 2 Report

The article “Biocontrol Potential of Chitin and Chitosan Extracted from 2 Black Soldier Fly Pupal Exuviae against Bacterial Wilt of Tomato” provides detailed information on the usefulness of Chitin and Chitosan Extracted from 2 Black Soldier Fly Pupal Exuviae against Bacterial Wilt of Tomato by Ralstonia solanacearum strains. So the authors make a valuable contribution to the Biocontrol of this plant disease with high yield losses in this valuable and important food in human diet.

This article is well structured and very well written and easy to perceive. So, the manuscript content, style and scope are well designed. Major changes are not required and only a brief revision of the below mentioned topics is sufficient. Therefore, I recommend accepting this article after minor revisions.

Please briefly review the following topics:

LINE 35: Ralstonia Solanacearum SHOULD BE Ralstonia solanacearum

LINE 39: in it being divided into four phylotypes SHOULD BE in it divided being into four phylotypes division

LINE 109: R.solanacearum SHOULD BE R. solanacearum

LINE 125: Pink short rod-shaped colonies SHOULD BE Pink short rod-shaped cells

LINE 134: A catalase oxidase test SHOULD BE A catalase test

Catalase (EC 1.11.1.6) is different from oxidase (EC 1.9.3.1) - cytochrome-c oxidase another enzyme that is searched in different way

LINE 151: R.solanacearum SHOULD BE R. solanacearum

LINE 154: Tran’s illuminator SHOULD BE Transilluminator

LINE 165: Tran’s illuminator SHOULD BE transilluminator

LINE 309: R.solanacearum SHOULD BE R. solanacearum

LINE 312: R.solanacearum SHOULD BE R. solanacearum

LINE 282: A catalase oxidase test SHOULD BE A catalase test

Catalase (EC 1.11.1.6) is different from oxidase (EC 1.9.3.1) - cytochrome-c oxidase another enzyme that is searched in different way

Author Response

Point by point suggestions:

Units are not properly formatted, and in several sections they are not separated from their numerical value, one (of multiple examples) is in line 175: “10mls”. In addition, in line 208 the authors use “Fifty microliters”, please keep the consistency and use adequate numbers and units.

corrected

The authors mention the use of a L-form glass spreader, how was it sterilized?

Described in line 218

Line 214, please consider replacing: “The diameter of zones of clearance around each disc (diameter of inhibition zone plus diameter of disc)” by: The inhibition zone (ZoI)”.

corrected

I failed to understand which was the concentration of antibiotics used, or which type was the material of the paper discs used. What type of paper? Pelase specify the material.

Specified in line 220

Line 135, “young bacterial culture”, please consider replacing the term “young” by the term “fresh”.

corrected

I beg your pardon but I failed to understand which type of statistical analysis were performed. Please create a subsection in the Material and Methods describing them.

Indicated in section 2.8

Equations should be referenced in the text and separated from the text body for a more clear understanding.

corrected

Finally, I did not found how the Black Soldier Fly Pupal were obtained. Were they commercially acquired? Cultured by the authors? Please clarify.

Clarified in line 203

Reviewer 3 Report

The manuscript entitled: “Biocontrol Potential of Chitin and Chitosan Extracted from Black Soldier Fly Pupal Exuviae against Bacterial Wilt of Tomato”, reference: microorganisms-1479818.

General comments:

The manuscript is clear, interesting, easy to follow and displays relevant insights. Nevertheless, its quality should to be improved. More importantly, in my opinion, the authors should clearly specify which is, on their opinion the best approach, the use of chitin or chitosan? This should explained and supported by the findings that the authors generated in this manuscript and it should be particularly addressed in the Conclusion section.

Point by point suggestions:

Units are not properly formatted, and in several sections they are not separated from their numerical value, one (of multiple examples) is in line 175: “10mls”. In addition, in line 208 the authors use “Fifty microliters”, please keep the consistency and use adequate numbers and units.

The authors mention the use of a L-form glass spreader, how was it sterilized?

Line 214, please consider replacing: “The diameter of zones of clearance around each disc (diameter of inhibition zone plus diameter of disc)” by: The inhibition zone (ZoI)”.

I failed to understand which was the concentration of antibiotics used, or which type was the material of the paper discs used. What type of paper? Pelase specify the material.

Line 135, “young bacterial culture”, please consider replacing the term “young” by the term “fresh”.

I beg your pardon but I failed to understand which type of statistical analysis were performed. Please create a subsection in the Material and Methods describing them.

Equations should be referenced in the text and separated from the text body for a more clear understanding.

Finally, I did not found how the Black Soldier Fly Pupal were obtained. Were they commercially acquired? Cultured by the authors? Please clarify.

Author Response

All comments are been submitted in the first 2 section